# `LSH-MoE`: Communication-efficient MoE Training via Locality-Sensitive Hashing

**Xiaonan Nie**[1]    **Qibin Liu**[1]    **Fangcheng Fu**[1]    **Shenhan Zhu**[1]    **Xupeng Miao**[2]
**Xiaoyang Li**[3]    **Yang Zhang**[3]    **Shouda Liu**[3]    **Bin Cui**[1]
[1]Peking University    [2]Purdue University    [3]ByteDance
[1]{xiaonan.nie,2101212782,ccchengff,shenhan.zhu,bin.cui}@pku.edu.cn
[2]xupeng@purdue.edu    [3]{lixiaoyang.x,zhangyang.elfin,liushouda}@bytedance.com

## Abstract

Larger transformer models always perform better on various tasks but require more costs to scale up the model size. To efficiently enlarge models, the mixture-of-experts (MoE) architecture is widely adopted, which consists of a gate network and a series of experts and keep the training cost constant by routing the input data to a fixed number of experts instead of all. In existing large-scale MoE training systems, experts would be distributed among different GPUs for parallelization, and thus input data requires additional all-to-all communications to access the target experts and conduct corresponding computations. However, upon evaluating the training process of three mainstream MoE models on commonly used GPU clusters, we found that the all-to-all communication ratio averaged around 45%, which significantly hinders the efficiency and scalability of training MoE models.

In this paper, we propose `LSH-MoE`, a communication-efficient MoE training framework using locality-sensitive hashing (LSH). We first present the problems of scaling MoE training in existing systems and highlight the potential of exploiting token similarity to facilitate data compression. Then, we introduce an efficient LSH-based compression technique, which utilizes the cross-polytope hashing for rapid clustering and implements a residual-based error compensation scheme to alleviate the adverse impact of compression. To verify the effectiveness of our methods, we conduct experiments on both language models (e.g., RoBERTa, GPT, and T5) and vision models (e.g., Swin) for pre-training and fine-tuning tasks. The results demonstrate that our method substantially outperforms its counterparts across different tasks by $1.28\times$ - $2.2\times$ of speedup.

## 1 Introduction

In recent years, large-scale pre-trained models have significantly advanced the performance of deep learning across various complex tasks, including computer vision [8, 20], natural language processing [3, 7, 28], and multi-modal learning [19]. Commonly referred to as foundation models, these pre-trained models are primarily built on Transformer architectures [34] and undergo extensive pre-training on large datasets, utilizing substantial GPU resources. OpenAI has validated the scaling law for large language models [15] and suggests that increasing the model's parameter size, the volume of training data, and the duration of training can significantly enhance the model's performance. However, this approach results in a considerable rise in training costs, making the development of foundation models extremely expensive.

---

Xiaonan Nie, Qibin Liu, Fangcheng Fu, Shenhan Zhu, and Bin Cui are with the School of Computer Science and Key Lab of High Confidence Software Technologies (MOE), Peking University. Bin Cui is also with the Institute of Computational Social Science, Peking University (Qingdao).

38th Conference on Neural Information Processing Systems (NeurIPS 2024).

To reduce the high computational costs, the sparse mixture-of-experts (MoE) architecture is often adopted, which comprises a sparse gate network and a series of expert networks. This architecture routes input data to only a subset of experts, resulting in sparse activation of the experts and thereby reducing the model's computational FLOPs (float point operations) as well as training costs. Prominent models such as Google's Switch-Transformer [9], ST-MoE [41], Meta's Hash Layer [31] and Mistral-AI's mixtral models [14] have successfully implemented this design, demonstrating improvements in both performance and efficiency with MoE models.

Meanwhile, effectively scaling the training of MoE models across hundreds or even thousands of GPUs remains a significant challenge. Researchers from Google have proposed the *expert parallelism* approach [17], which replicates the gating network on each GPUs and distributes different experts across multiple GPUs for parallel processing. Specifically, each input token is initially processed by the gating network to select the appropriate expert, after which it is routed to the designated experts via peer-to-peer (P2P) network communication. Once the designated experts complete their computation, the token is returned to the original GPU for further processing through an additional P2P communication. Since each GPU typically needs to exchange data with many other GPUs, these P2P transmissions results in an all-to-all communication pattern. Moreover, because the computation of the expert network relies on the outcomes of these communications, the communications cannot be effectively overlapped with ongoing computations. This dependency creates a significant performance bottleneck in model training across most commonly used GPU clusters. We conducted experiments on three widely-used MoE models, including RoBERTa-MoE, GPT-MoE and Swin-MoE, on four A100 servers, each with a cross-machine bandwidth of 200Gb/s. The results, as shown in Figure 3, reveal that the time cost of all-to-all communication constitutes an average of $45\%$ and can reach up to $67\%$ of the total model training time.

Existing methods to improve distributed MoE training on bandwidth-limited clusters tackle communication challenges in various ways. TA-MoE [4] reduces cross-machine communication by adjusting the gating network to favor experts on the same server, while Pre-gated MoE [13] reduces dependency between communication and computation through a pre-gating mechanism that plans token routing in advance. However, both approaches require modifications to the gating mechanism and model structure, limiting their universal applicability. DeepSpeed-MoE [29] introduces PR-MoE, which selects one expert plus a shared expert, halving the all-to-all communication load. SCoMoE [40] organizes all-to-all communication by structuring data transfers along different dimensions and controlling data volumes across network levels, and also clusters tokens to improve routing. However, none of these works consider reducing the All-to-All communication volume in MoE training by compressing the forward activations. Therefore, they can be intergrated with our method for further improvement.

In this paper, we present `LSH-MoE`, a communication-efficient MoE training framework that leverages locality-sensitive hashing to group similar tokens. Our key contributions are as follows:

- We begin by identifying key challenges in scaling MoE training in existing systems, noting that all-to-all communication constitutes an average of $45\%$ of the total training time. Additionally, we investigate the potential of using token similarity to facilitate data compression to reduce communication costs.

- We propose an efficient LSH-based compression technique that employs cross-polytope hashing for rapid clustering. This approach transmits only the clustering centroids, significantly reducing communication costs. To further enhance accuracy, we implement a residual-based error compensation scheme to mitigate the negative effects of compression.

- Through extensive experiments with language models (RoBERTa-MoE, GPT-MoE, and T5-MoE) and vision models (Swin-MoE), across both pre-training and fine-tuning tasks, we demonstrate that our method maintains model quality while achieving a speedup of $1.28\times$ - $2.2\times$ in end-to-end training time.

## 2 Background

### 2.1 Mixtures-of-Expert Architecture

To enhance the training efficiency of Transformer models, William et al. (2022) [9] introduced an innovative paradigm, the sparse mixture-of-eexperts (MoE) architecture, illustrated in Figure 1.

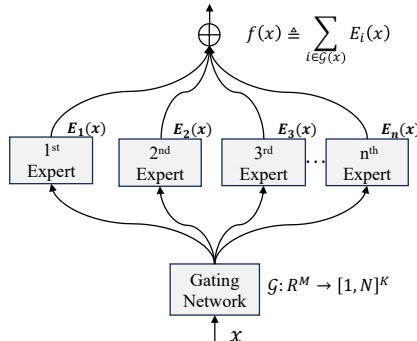

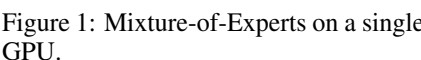

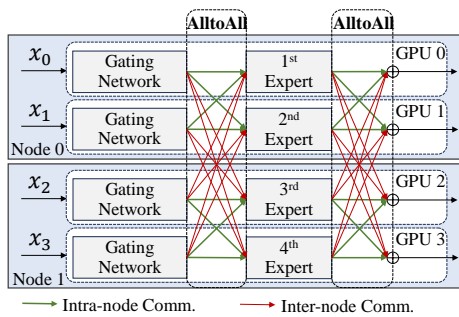

Figure 1: Mixture-of-Experts on a single GPU.

Figure 2: Training Mixture-of-Experts on multiple GPUs as expert parallelism.

This architecture effectively balances parameter capacity and training costs, and comprises two key components: an *expert network* ($\mathcal{E}$) and a *sparse gate network* ($\mathcal{G}$). It is evident that MoE models, with an equal number of active parameters per input, can significantly surpass the performance of dense models. This breakthrough has also catalyzed further research and their application across various industries, as highlighted by numerous subsequent studies [5, 14, 22, 23, 25, 30, 39].

The *expert network* $\mathcal{E}$ is composed of multiple specialized and separate networks, commonly referred to as *experts*, denoted as $\{E_i\}_{i=1}^{N}$, where $N$ represents the number of experts. Additionally, $E_i(x)$ denotes the output produced when the input $x$ is processed by the $i$-th expert. Each expert is trained to excel in a specific sub-task, such as in multi-task learning, or to handle specific segments of data, as seen in language modeling and multi-modal learning, thereby increasing the overall model capacity. In foundational models, the MoE layer often serves as a substitute for the traditional feed-forward network (FFN) layer. Within each MoE layer, each FFN function works as an individual expert, significantly enhancing the model's capability to process diverse and complex data inputs.

The *gating network* $\mathcal{G}$ plays a crucial role in the sparse MoE architecture. For example, in a $K$-way gated MoE system, the gating network outputs a set of integers as Equation 1 to determine which experts should be activated. This decision is based on the characteristics of the input itself, allowing for a dynamic and efficient allocation of computational resources. By only processing each input token with a selected subset of the expert network, the MoE model achieves computation sparsity, effectively decoupling parameter capacity from training costs.

$$\mathcal{G} : \mathbb{R}^M \to [1, N]^K \tag{1}$$

Through the integration of multiple specialized experts, as described by Equation 2, the sparse MoE model is capable of delivering more accurate and efficient predictions as $f(x)$. This is achieved by leveraging the specialized knowledge embedded within each expert, combined with the strategic input allocation managed by the gating network.

$$f(x) \triangleq \sum_{i \in \mathcal{G}(x)} E_i(x) \tag{2}$$

While MoE's primary advantage is decoupling parameter capacity from network cost, a key challenge lies in learning the gating parameters effectively, as the output's sparsity makes it non-differentiable. Consequently, much of the research in the MoE field has centered on developing methods for learning gating functions. These methods fall into three main categories, as outlined in [6]: routing via learnable weighting [9, 24, 30], deterministic hash routing [31], and reinforcement learning-based routing [2, 32, 33]. These approaches primarily differ in the design of the gating network $\mathcal{G}$ rather than the expert network $\mathcal{E}$, and therefore all encounter similar scaling challenges.

## 2.2 Challenges of Scaling MoE Model Training

While MoE models were initially developed to facilitate efficient scaling during training, deploying these large-scale models in practical GPU-intensive environments poses significant challenges in

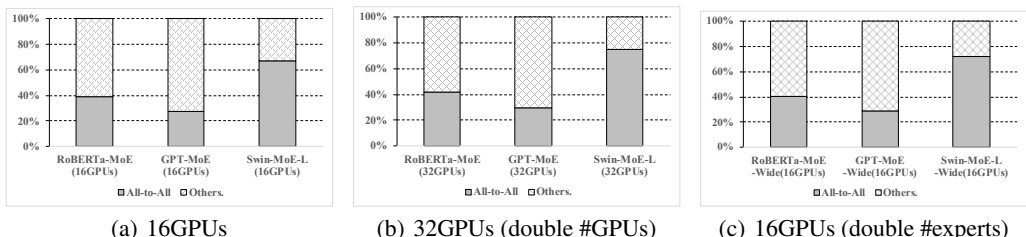

|  | (a) 16GPUs | (b) 32GPUs (double #GPUs) | (c) 16GPUs (double #experts) |

Figure 3: Proportion of all-to-all communication time relative to total training duration across different configurations: scaling the number of training servers (Figure 3(b)) and scaling the parameter size of models (Figure 3(c)).

distributed computing. Specifically, the MoE layer harbors a considerably higher number of parameters and requires additional memory, yet it maintains almost the same computational demands as the dense layer. This leads to a unique *compute density* — defined as the ratio of the layer's FLOPs (Floating Point Operations) to its number of parameters. Therefore, traditional parallelism methods such as *tensor parallelism* and *pipeline parallelism* are insufficient for achieving effective parallelism in the scenarios of MoE training.

To improve the efficiency and scalability of training large-scale MoE models, *expert parallelism* [17] has been introduced as a specialized model parallelism strategy. This approach distributes experts within an MoE layer across multiple GPUs, while leveraging data parallelism for replicating non-MoE layers, thus efficiently managing the training workload of MoE models. The workflow of distributed training for an MoE layer is depicted in Figure 2. Once the target expert for each token is determined, an all-to-all communication process is triggered to distribute tokens to their corresponding target experts for computations, denoted as $E_i(x)$. Subsequently, another round of all-to-all communication is executed to gather the outputs from all experts, which produces the MoE layer's output (represented as $f(x)$, Equation 2). Subsequent operations involve executing the data-parallel non-MoE layers.

We first profiled the training process of three popular MoE models employing expert parallelism (detailed in Table 1) on a cluster comprised of four A100 machines, each equipped with an interconnect RDMA bandwidth of 200Gb/s. The proportion of all-to-all communication time relative to the total training duration is illustrated in Figure 3(a). We then double the number of machines, and the number of experts to increase the model scale. The results are shown in Figure 3(b) and 3(c), respectively. Our findings reveal that all-to-all communication accounted for a substantial portion of the total time: approximately 30% in GPT-MoE (15B), 40% in RoBERTa-MoE, and 70% in Swin-MoE-L. And this overhead remains nearly constant in larger models and at larger machine scales. These results highlight a significant bottleneck that hampers the scalability of the training process. Consequently, the duration of all-to-all communication substantially constrains training with expert parallelism, leading to reduced overall throughput and limiting the potential to scale up the number of experts effectively.

## 2.3 Locality-Sensitive Hashing Algorithms

Locality-Sensitive Hashing (LSH) is a probabilistic method primarily used to approximate nearest neighbor search in high-dimensional spaces, which reduces the dimensionality of data by mapping similar data to the same "buckets" with high probability using hash functions. This approach offers a substantial reduction in computational complexity, particularly beneficial for large-scale data applications. The key operations in LSH including:

**Mapping Data into Buckets:** The core of LSH is a family of hash functions that maximize the probability of nearby points in the original space staying close in the hashed space, while distant points are likely to end up in different buckets. Each hash function $h$ is characterized by the property: $P[h(x) = h(y)] = 1 - d(x,y)/D$, where $d(x,y)$ is the distance between points $x$ and $y$, and $D$ denotes the diameter of the space. To map similar data into the same bucket, multiple hash functions from this family are selected based on the specific attributes of the data (e.g., Euclidean distance, cosine similarity) and the desired granularity of the buckets. Data points are then hashed by these

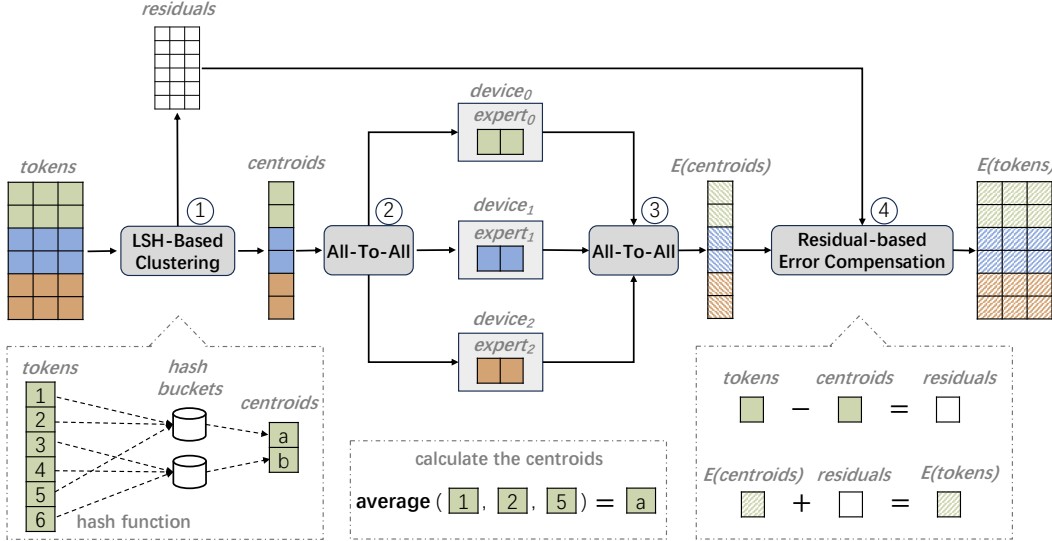

Figure 5: Schematic of MoE training with Locality-Sensitive Hashing (LSH-MoE).

functions, and each point is assigned to buckets according to its hash values, effectively categorizing similar items together for clustering.

**Calculating Cluster Centroids:** By grouping data points into buckets as determined by their hash values, data points are effectively clustered. Each bucket represents a cluster of data points and the centroid of each cluster is then calculated as the mean of all points within that cluster, formulated as $C_j = \frac{1}{n} \sum_{i=1}^{n_j} x_i$, where $C_j$ is the centroid of the j-th bucket, $n_j$ is the number of points in the j-th bucket, and $x_i$ are the data points in the bucket.

## 3 Methodology

### 3.1 The Motivation of Token Similarity

To explore the potential optimization for all-to-all communications in MoE training, we conducted an in-depth analysis of the data involved in these all-to-all communications, identifying a high degree of similarity, termed *token similarity*. Specifically, we applied *Principal Component Analysis* (PCA) to reduce the dimensionality of the input tokens of all-to-all communications and observed a distinct clustering phenomenon, as illustrated in the Figure 4. Our analysis suggests that the observed similarity among tokens may stem from two primary factors:

- *Data Related Influences*: The similarity is partially due to the nature of real-world data, which often adheres to Zipf's Law [18]. This results in a skewed distribution, with certain data elements appear more frequently than others.

- *Model Structure Related Influences*: The design of Transformer architecture [34], especially its attention mechanisms, significantly impacts token similarity. In models like BERT [7], attention layers are designed to capture and integrate context information across tokens, thus homogenizing token representations and emphasizing their shared semantic relationships at the sentence level.

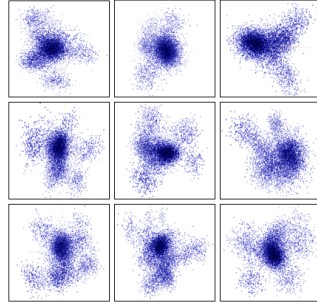

Figure 4: Principal Component Analysis (PCA) Visualization of input tokens involved in all-to-all communication.

## 3.2 LSH-MoE

Motivated by the *Token Similarity* observed in Section 3.1, we introduce LSH-MoE, a novel MoE training framework that integrates locality-sensitive hashing (LSH) for rapid clustering of input tokens. Our method transmits only the clustering centroids, significantly reducing communication volumes. To counteract the negative effects of compression, we also implement a residual-based error compensation scheme.

As depicted in Figure 5, LSH-MoE initially employs (1) an LSH-based clustering method to compress *tokens* into *centriods* for subsequent processing, effectively reducing communication overhead. It then sequentially executes (2) all-to-all communication, expert computation, and another (3) all-to-all communication to produce the processed outputs *E(centriods)*. Finally, it introduces (4) a residual-based error compensation method to approximate the expert-processed results *E(tokens)*, by integrating *E(centriods)* with *residuals*. Meanwhile, we also outline the workflow of our LSH-MoE framework in the Algorithm 1 of Appendix A.1. The key components of our LSH-MoE framework includes **an efficient LSH-based clustering algorithm** for rapid processing and **an residual-based error compensation scheme** to minimize quality degradation.

**Efficient LSH-based Clustering Algorithm.**    Since the data to be compressed (the input data for all-to-all communication) is generated dynamically and in real time, pre-compressing it or overlapping compression time with other processing tasks is not feasible. Consequently, selecting an efficient online compression algorithm is crucial. Traditional clustering algorithms, such as K-Means, often encounter computational challenges and efficiency limitations. Locality-sensitive hashing (LSH) address these issues by hashing similar data points into the same buckets, enabling faster similarity detection in high-dimensional spaces.

Numerous LSH algorithms have been developed, each employing a unique hashing approach for mapping data onto buckets. We conducted experiments to evaluate several popular hashing algorithms, including *cross-polytope hashing* and *spherical hashing*. Based on our evaluations in Section 4.5, we selected *cross-polytope hashing* as the optimal algorithm for our application. *Cross-polytope hashing* stands out for its method of mapping input vectors to the nearest vertex on a cross-polytope. This process is facilitated by applying randomly rotated cross-polytopes, which effectively segment the surface of the unit sphere. The algorithm can be mathematically represented as follows:

$$LSH(\mathbf{x}) = \text{argmax}_{i \in \{\pm 1, \pm 2, \ldots, \pm d\}} |\mathbf{R}\mathbf{x}|_i \tag{3}$$

where $\mathbf{R}$ is a random rotation matrix, $d$ is the dimensionality of the space, and $|\mathbf{R}\mathbf{x}|_i$ denotes the absolute value of the $i$-th component of the rotated vector $\mathbf{R}\mathbf{x}$.

This formula encapsulates how the input vector $x$ is transformed by the rotation matrix $R$ and then mapped to the nearest vertex of the cross-polytope by selecting the dimension $i$ that maximizes the absolute value of the components of $Rx$. This method effectively segments the high-dimensional space and enhances the clustering efficiency by rapidly identifying similar data points.

**Residual-based Error Compensation Scheme.**    In our LSH-MoE framework, we compress the intermediate activation values within the model network. Unlike gradient compression, this process does not tolerate errors well. Therefore, it is essential to minimize compression-induced errors to ensure minimal impact on model performance. To address this, we implement a novel residual-based gradient compensation strategy, outlined as follows:

1. We first capture the residual for each data point relative to its cluster centroid, defined by the equation:
$$\Delta \text{cluster}_j \leftarrow \{x - \overline{\text{cluster}}_j \mid x \in \text{cluster}_j\}. \tag{4}$$

2. After the expert network computes outputs for the cluster centers, the final step is to restore the processed result for each token by adding back the previously recorded residual:
$$Y_{ij} \leftarrow \{E(\overline{\text{cluster}}_j) + \Delta \text{Cluster}_{jk} \mid k = 1, 2, \ldots, N_j\}. \tag{5}$$

This error compensation scheme effectively mitigates potential accuracy loss caused by data compression in all-to-all communication, ensuring the fidelity and robustness of the LSH-MoE framework. The experimental results in Section 4 show that implementing this compensation mechanism enables

Table 1: Models for evaluation, where "-" indicates that the values are different across layers.

| Model | #Layer | $d_{model}$ | $d_{ffn}$ | #Experts | #Params. (MoE) | #Params. (Total) |
|---|---|---|---|---|---|---|
| RoBERTa-MoE | 12 | 768 | 3072 | 16 | 302M | 394M |
| T5-MoE | 16 | 1024 | 16384 | 16 | 8594M | 9288M |
| GPT-MoE (15B) | 12 | 768 | 3072 | 512 | 14507M | 14629M |
| GPT-MoE (52B) | 24 | 1024 | 4096 | 512 | 51539M | 51740M |
| Swin-MoE-L | 24 | - | - | 32 | - | 946M |

the model trained with LSH-MoE to achieve an accuracy comparable to that of a model trained without compression. This outcome highlights the effectiveness of our proposed error compensation strategy in preserving model performance despite the challenges posed by data compression in all-to-all communication.

### 3.3 Scalability Analysis of LSH-MoE

To effectively demonstrate the scalability of our approach, particularly in terms of its applicability to both larger models and larger computational clusters, we conducted a theoretical analysis. This analysis primarily focuses on the **computation overhead** and the communication costs associated with Mixture of Experts (MoE), specifically considering **all-to-all communication overhead**. We derived the ratio of communication time to computation time, highlighting how this ratio evolves as both the scale of the servers and the model size increase. This relationship is crucial for understanding scalability and can be formally expressed as follows:

$$\frac{T_{all\_to\_all}}{T_{compute}} = \frac{\text{FLOPs}}{6B_{inter}} \times \frac{k}{1 + 2k} \times \frac{w - 1}{wh} \tag{6}$$

where $k$ represents the number of experts activated per token, FLOPs and $B_{inter}$ denote the GPU's computation ability and the network performance, $w$ is the number of GPU servers, and $h$ is the hidden size of model. Notably, the first term, $\frac{\text{FLOPs}}{6B_{inter}}$, remains constant under fixed hardware conditions. Additionally, scaling MoE models typically emphasizes increasing the number of layers and experts, while the growth in hidden size ($h$) tends to be gradual, as seen in models like Switch-Transformer [9]. Consequently, when both the model scale and the number of training servers grow, the proportion of all-to-all communication time remains nearly unchanged. This insight underpins the scalability of the LSH-MoE method, demonstrating its robustness in larger-scale settings and supporting its potential in future large-scale applications. For a detailed derivation, please refer to Appendix A.2.

## 4 Experiment

### 4.1 Implementation

Our LSH-MoE comprises a data compression/restoration component and a communication component. We utilize PyTorch 1.11 for developing the LSH clustering and NCCL for implementing the communication. Additionally, our method is framework-independent and can be easily applied to other MoE training frameworks such as Hetu-MoE [21, 26], DeepSpeed-MoE [29], and Tutel [12].

### 4.2 Benchmarks and Datasets

Our evaluations are conducted by scaling pre-trained models equipped with MoE architecture across various application domains. This includes models like RoBERTa-MoE, T5-MoE and GPT-MoE in natural language processing (NLP), as well as Swin-MoE in computer vision (CV). Among these models, RoBERTa-MoE and T5-MoE are evaluated on pre-training task, while GPT-MoE and Swin-MoE undergo fine-tuning evaluation based on their official open-sourced model checkpoints [1] [2]. We also evaluated the zero-shot accuracy of the pre-trained T5-MoE. Model configurations are detailed in Table 1.

---

[1] https://github.com/facebookresearch/fairseq/tree/main/examples/moe_lm
[2] https://github.com/microsoft/Swin-Transformer/blob/main/MODELHUB.md

The RoBERTa-MoE model is pre-trained with masked language modeling tasks on a combined dataset, which includes BooksCorpus ($\sim$ 800M words) and English Wikipedia ($\sim$ 2,500M words). This dataset is tokenized using a tokenizer with a vocabulary size of 50,257. To assess the impact of our MoE method in compressing all-to-all communication on large model training, the T5-MoE model, which is with about 10B parameters, is pre-trained on an industry dataset ($\sim$ 500M words) using a span-masked language modeling task. In addition to pre-training tasks, we further evaluate our work on fine-tuning tasks. To be specific, we fine-tune two open-sourced models, including the language model GPT-MoE on the General Language Understanding Evaluation (GLUE) benchmark and the vision model Swin-MoE on the ImageNet classification benchmark.

### 4.3  Software and Hardware Environments

To thoroughly evaluate the effectiveness of our method, we conducted experiments on two clusters, V100 cluster and A100 cluster. Additionally, to ensure consistency in software versions, we performed experiments on both machines using the same docker image.

**Software Environment.** Our experiments were conducted using a docker image built upon the official NVIDIA GPU containers, which includes Ubuntu 20.04, CUDA 11.3, cuDNN 8.2.0, and NCCL 2.12.7, accessible at NVIDIA GPU Containers [3].

**V100 Cluster.** The first hardware environment includes two servers, each outfitted with eight NVIDIA V100 (32GB) GPUs. Within each server, GPUs are interconnected using NVLink 2.0 technology. The servers are interconnected via an RDMA NIC, providing a network bandwidth of 100 Gbps.

**A100 Cluster.** The second hardware environment consists of four servers, each equipped with eight NVIDIA A100 (40GB) GPUs. Within these servers, GPUs utilize NVLink 3.0 technology for interconnection. The servers are linked through two RDMA NICs, enhancing the network bandwidth to 200 Gbps.

We allocated the experiments involving RoBERTa-MoE and GPT-MoE to the V100 cluster, while T5-MoE and Swin-MoE were tested on the A100 cluster. This setup allowed us to effectively compare the performance impacts across different hardware configurations.

### 4.4  Overall Performance

In general, to evaluate our LSH-MoE training approach, which compresses communication data, there are two crucial questions:

1. Does the LSH-MoE method enable normal model convergence, and is there a risk of increased loss variability during this process, potentially leading to instability in training?

2. Might the implementation of the LSH-MoE method adversely affect the model's performance on downstream benchmarks?

Therefore, we conducted experiments focusing on both **Convergence Performance** and **Benchmark Performance** to validate the effectiveness of our method. In this section, due to the necessity of selecting several hyperparameters for LSH, such as the type of hash function and the quantity of hash functions, we have opted for the cross-polytope hash function based on empirical evaluation, setting the number of hash functions at 6. A detailed examination of the effects stemming from variations in these parameters will be methodically addressed in the upcoming ablation study (Section 4.5).

**Convergence Performance.** We pre-trained the RoBERTa-MoE and T5-MoE using open-source datasets and industrial datasets, respectively. In our approach, we substitute the FFN (Feed-Forward Network) layer with an MoE (Mixture of Experts) layer in alternating layers, as detailed in Section 4.2. We meticulously tracked the time required to achieve equivalent model performance levels (perplexity) during training, as depicted in Figure 6. The results indicate a significant acceleration in training convergence when employing the LSH-MoE method: $1.6\times$ faster for RoBERTa-MoE and $2.2\times$ faster for T5-MoE, compared to the original models' convergence rates. Furthermore, we investigated the role of error compensation in this process. Our findings reveal that omitting error compensation in the LSH-MoE model led to a 0.3 point increase in perplexity, given the same training duration. This observation underscores the efficacy of the error compensation algorithm.

---

[3] https://catalog.ngc.nvidia.com/orgs/nvidia/containers/pytorch

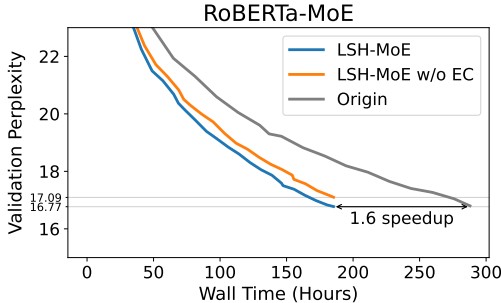
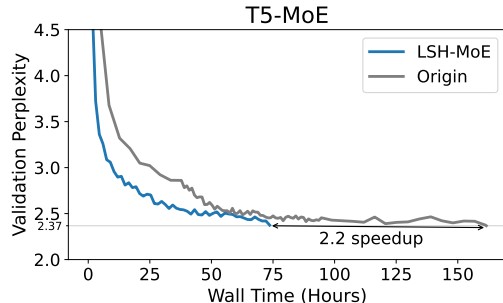

Figure 6: Comparative analysis of convergence performance. This includes a comparison between the original models, LSH-MoE without Error Compensation, and LSH-MoE implementations. The perplexity curves are applied 1D Gaussian smoothing with $\sigma = 0.5$.

Table 2: Evaluation of LSH-MoE on the GLUE benchmark.

| Dataset | GPT-MoE (15B) | | | GPT-MoE (52B) | | | T5-MoE (10B) | |
| | Origin | Ours | Speed | Origin | Ours | Speed | Origin | Ours |
| --- | --- | --- | --- | --- | --- | --- | --- | --- |
| SST-2 | 93.8% | 93.8% | 1.3× | 94.5% | 94.3% | 1.4× | 51.6% | 50.9% |
| MNLI | 82.8% | 82.7% | 1.4× | 84.1% | 84.3% | 1.4× | 52.6% | 52.1% |
| QNLI | 86.6% | 86.7% | 1.3× | 90.2% | 90.0% | 1.5× | 49.5% | 50.0% |
| QQP | 88.8% | 88.7% | 1.3× | 88.9% | 88.9% | 1.2× | - | - |
| MRPC | 71.3% | 71.1% | 1.3× | 76.3% | 76.1% | 1.3× | - | - |
| COLA | 72.3% | 72.4% | 1.4× | 73.5% | 73.8% | 1.5× | - | - |

Table 3: Results of fine-tuning Swin-MoE on the ImageNet-1K dataset.

| | Origin | Ours |
| --- | --- | --- |
| Top-1 Acc. ↑ | 84.7% | 84.5% |
| Top-5 Acc. ↑ | 97.0% | 97.1% |
| Compression Rate | — | 11.7% |
| Sample/s | 184.3 | 236.6 |

**Benchmark Performance.** To better validate the performance of LSH-MoE on downstream tasks, we fine-tuned the GPT-MoE and Swin-MoE on different datasets using open-source model checkpoints, and evaluated zero-shot performance of our internal pre-trained T5-MoE model, adhering to their original architectural designs that incorporate Top-2 gating, as detailed in [1, 12, 17].

We first utilized the LSH-MoE method for fine-tuning the GPT-MoE of two model scales (i.e. 15B and 52B) on the GLUE benchmark, yielding impressive outcomes. As detailed in Table 2, the implementation of the LSH-MoE method substantially reduced communication overhead while maintaining nearly the same level of accuracy. This strategy resulted in a significant performance boost, achieving an acceleration rate ranging from 1.2× to 1.5×. The results also demonstrate that as the parameter size of MoE models increases, LSH-MoE continues to achieve significant improvements without compromising model accuracy. Additionally, we report the zero-shot accuracy of the pre-trained T5-MoE, showing that the T5-MoE models trained with LSH-MoE achieved accuracy comparable to standard T5 models, confirming LSH-MoE's efficacy in pretraining. Because the limited number of tokens in the pre-trained dataset and its out-of-domain nature compared to the GLUE evaluation data, the zero-shot performance metrics are relatively low.

Furthermore, our evaluation of the LSH-MoE method in fine-tuning the Swin-MoE on the ImageNet-1K dataset demonstrated noteworthy efficiency. We achieved a communication compression rate of 11.7%, which led to a 1.28× increase in acceleration, as reported in Table 3. Notably, this was accomplished while preserving almost the same level of accuracy.

## 4.5 Ablation Study

To study the impact of the quantity and types of hash functions, we conducted ablation experiments by fine-tuning the GPT-MoE (15B) model on the MNLI and SST-2 datasets in the GLUE benchmark.

**Impact of the Quantity of Hash Functions.** We controlled the number of hash functions to indirectly adjust the LSH compression rate, exploring its effect on model performance. Specifically, we utilized 2, 4, 6, 8, and 10 hash functions. As shown in the middle sub-figure in Figure 7, we observe that an increase in the number of hash functions enlarges the number of buckets, enhances data distinction and, consequently, the compression rate. Besides, we indicate from the left sub-figure in Figure 7, that more hash functions leads to improved model convergence quality and worse compression rate.

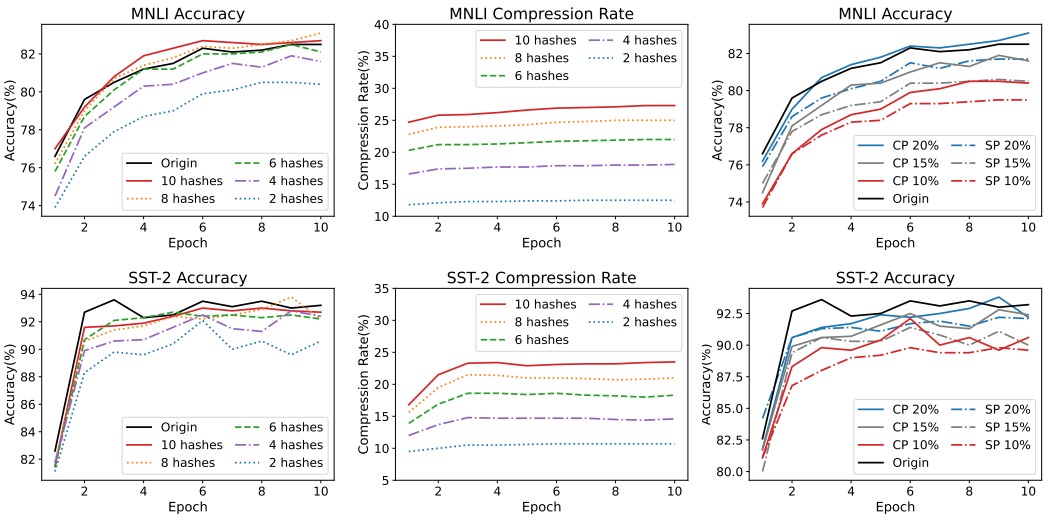

Figure 7: An in-depth analysis of the compression rate and the model performance by adjusting the **quantity** and **types** of hash functions. The left and middle sub-figures are results for diverse quantities of hash functions. The right sub-figure is the result for diverse types of hash functions (CP for cross-polytope and SP for spherical) with different compression rates (20%, 15%, 10%).

Importantly, our results indicate that a compression rate of approximately 20% (achieved with about 6 hash functions) is optimal for maintaining nearly identical convergence as uncompressed models. Therefore, we choose 6 as the default number of hash functions in Section 4.4.

**Impact of the Types of Hash Functions.** We further explored the impact of the types of hash functions with Cross-Polytope Hashing (CP) and Spherical-Plane Hashing (SP). The outcomes are illustrated in the right sub-figure in Figure 7. CP generally achieves better convergence than SP at the same compression rate. This is attributable to CP's ability to more effectively handle a variety of complex data patterns. CP encodes data based on an n-dimensional cross-polytope, while SP relies on the geometric relationships between spheres and planes. Thus, CP is more generalizable across a variety of complex data patterns while SP performs better with data that has spherical distribution characteristics. Other works (e.g. Reformer [16]) also use CP to leverage the sparsity of attention mechanisms. Therefore, we finally choose **cross-polytope hashing** as the default type of hash functions in Section 4.4.

# 5   Conclusion

Our study tackled the latency challenges inherent in training sparse-gated Mixture-of-Experts (MoE) models with our innovative LSH-MoE framework. Utilizing locality-sensitive hashing to harness token similarities, our approach significantly reduces communication overhead. The integration of a residual-based error compensation scheme further preserves model integrity under compression. Empirical tests across various models, including RoBERTa, GPT, T5, and Swin, showcase LSH-MoE's capability to accelerate both pre-training and fine-tuning phases by up to $2.2\times$, paving the way for efficient and scalable MoE applications in real-world settings.

# 6   Limitations

At the current stage, our work only considers MoE models. Nevertheless, we want to clarify that MoE models are also a mainstream class of deep learning models that are increasingly adopted due to rising model computational demands and training costs, such as Mixtral-7Bx8MoE, DeepSeek-MoE, and GPT-4. Hence, accelerating MoE training is indeed a critical direction. Additionally, the core of our work leverages data redundancy, which is also presented in non-MoE model training. We hope our observations and utilization of data redundancy can inspire more refined work in optimizing training for non-MoE models as well.

## Acknowledgments and Disclosure of Funding

This work is supported by Beijing Natural Science Foundation (4244080), National Natural Science Foundation of China (U22B2037, U23B2048), Beijing Municipal Science and Technology Project (Z231100010323002), China National Postdoctoral Program for Innovative Talents (BX20230012), China Postdoctoral Science Foundation (2024M750103), research grant No. SH-2024JK29, ByteDance-PKU joint program (PJ20230202900065), and High-performance Computing Platform of Peking University. Fangcheng Fu and Bin Cui are the corresponding authors.

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

---

**Algorithm 1** Training MoE models using our LSH-MoE framework

---
**Input**: $X$: sequence of tokens
**Output**: $\{Y_{ij} \mid 1 \leq i \leq n, 1 \leq j \leq m\}$, where $Y_{ij}$ is the output for tokens in the $j$-th cluster assigned to the $i$-th expert

1: **function** MOE_LAYER_WITH_LSH($X$)
2:      Calculate the token-to-expert mapping $\zeta$ using the `gating network`;
3:      Dispatch $X$ into $\{X_i \mid i = 1, 2, \ldots, n\}$ based on $\zeta$; // $X_i$ are tokens assigned to the $i$-th expert
4:      **for** $i \leftarrow 1, 2, \ldots, n$ **do**
5:          $IDX_i \leftarrow LSH(X_i)$; // Get the LSH bucket for each token
6:          Divide $X_i$ into $\{\text{cluster}_j \mid j = 1, 2, \ldots, m\}$ based on $IDX$;
7:          **for** $j \leftarrow 1, 2, \ldots, m$ **do**
8:              $\overline{\text{cluster}}_j \leftarrow \text{Mean}(\text{cluster}_j)$; // Get the centroids for each cluster
9:              $\Delta\text{cluster}_j \leftarrow \{x - \overline{\text{cluster}}_j \mid x \in \text{cluster}_j\}$; // Get the difference between each token and its cluster centroids
10:          $C_i \leftarrow \{\overline{\text{cluster}}_j \mid j = 1, 2, \ldots, m\}$;
11:          $\Delta X_i \leftarrow \bigcup_{j=1}^{m} \Delta\text{cluster}_j$;
12:      $C \leftarrow \{C_i \mid i = 1, 2, \ldots, n\}$;
13:      $Input \leftarrow$ all-to-all($C$); // Transmit the cluster centroids through all-to-all
14:      $Output \leftarrow \text{Expert}(Input)$; // Perform computations on centroids
15:      $E(C) \leftarrow$ all-to-all($Output$); // Transmit the results back through all-to-all
16:      **for** $(i, j) \leftarrow (1, 2, \ldots, n) \times (1, 2, \ldots, m)$ **do**
17:          $Y_{ij} \leftarrow \{E(\overline{\text{cluster}}_j) + \Delta\text{Cluster}_{jk} \mid k = 1, 2, \ldots, N_j\}$; // Apply the residual-based error compensation scheme
18:      **return** $\{Y\}$.

---

# A   Appendix

## A.1   Framework of LSH-MoE

As illustrated in Algorithm 1, our LSH-MoE training process begins by dispatching each input token in $X$ to its designated expert based on the gating network (Line 2-3). It then utilizes locality-sensitive hashing to cluster tokens into groups, calculating the centroid for each cluster to represent the mean of its tokens, and recording the differences between each token and its centroid for later error compensation (Lines 4-11). These centroids are subsequently transmitted to the experts via all-to-all communication for processing and their results are sent back in a similarly manner (Line 13-15). Finally, a residual-based error compensation is applied to determine the output of the MoE layer (Lines 16-18). This method effectively minimizes the communication load, thus improving the scalability and efficiency of MoE model training.

## A.2   Scalability Analysis

First of all, we want to highlight that as the scale of both models and machines increases, the proportion of all-to-all communication time relative to the total time remains nearly constant. This consistency suggests that the LSH-MoE method remains effective even at larger scales. We will now present our derivation step by step, using the notations listed in Table 4.

**Formulate all-to-all communication.** For any given training server, the amount of tokens (i.e. $m$) communicated with any other GPU node can be expressed as $m = n \times k/w$. Similarly, the volume of communication within the same GPU node is also equal to $m = n \times k/w$. Consequently, the time required for all-to-all communication during model training can be modeled as follows, with each layer involving two instances for the forward pass and two for the backward pass:

$$T_{all\_to\_all} = 4 \times l \times \left( \frac{m \times h}{B_{intra}} + \frac{m \times h \times (w - 1)}{B_{inter}} \right) \approx 4l \times \frac{nk}{w} \times \frac{h(w - 1)}{B_{inter}}. \quad (7)$$

**Formulate model computation.** Based on the derivation in [27], for a standard decoder model, given the number of layers $l$, and the hidden size $h$ of the model, the activated parameter count per token

Table 4: Notations used in scalability analysis.

| Notation | Description |
| --- | --- |
| $n$ | The number of tokens processed per GPU |
| $m$ | The number of tokens communicated between two training servers |
| $k$ | The number of experts activated per token |
| $h$ | Hidden size for each token |
| $l$ | The number of layers for the model |
| $w$ | The number of training servers |
| $B_{intra}$ | The intra-machine network bandwidth |
| $B_{inter}$ | The inter-machine network bandwidth |

can be formalized as #ActivatedParams. $= 4(1 + 2k)lh^2$. According to the theory in the appendix of the GPT-3 paper [3], the computation time per GPU can be formalized as $T_{compute}$, where FLOPs represents the computation ability of GPU.

$$T_{compute} = 6 \times \text{\#tokens} \times \frac{\text{\#ActivatedParams.}}{\text{FLOPs}} = \frac{24(1 + 2k)nlh^2}{\text{FLOPs}}. \tag{8}$$

**Formulate all-to-all communication / computation.** Therefore, as the machine scale ($w$) and model scale ($l$ and $h$) increase, the ratio of computation time to communication time can be formalized as:

$$\frac{T_{all\_to\_all}}{T_{compute}} = \frac{4l \times \frac{nk}{w} \times \frac{h(w-1)}{B_{inter}}}{(24(1 + 2k)nlh^2)/\text{FLOPs}} = \frac{\text{FLOPs}}{6B_{inter}} \times \frac{k}{1 + 2k} \times \frac{w - 1}{wh}, \tag{9}$$

where the first term $\frac{\text{FLOPs}}{6B_{inter}}$ is constant. As MoE models scale up, the emphasis is generally placed on increasing the number of layers and experts, with a more gradual increase in hidden size (e.g. Switch-Transformer [9]). Consequently, the proportion of communication time remains significant as both the model size and the number of servers increase. These observations and theoretical proofs underscore the sustained effectiveness of the LSH-MoE method in larger environments, thus reinforcing its scalability and applicability for future advancements.

