# OpenReview forum: "LSH-MoE: Communication-efficient MoE Training via Locality-Sensitive Hashing"
_NeurIPS.cc/2024/Conference — NeurIPS 2024 poster_

### Official Review · Reviewer_4XaA · 2024-07-08

**Soundness:** 2
**Presentation:** 3
**Contribution:** 2
**Rating:** 6
**Confidence:** 3

**Summary:**

The authors focus on the communication overhead in large-scale MoE training, specifically under the expert parallel plus the data parallel regime. They reduce the communication workload by transmitting only the clustering centroids, which are calculated on the fly using LSH functions. To reduce the compression-induced error, the authors also propose a residual-based error compensation scheme. The authors perform evaluations using 4 different architectures  in both language and vision, and for both pertaining and finetuning tasks.

**Strengths:**

1. This paper is primarily well-written, and the algorithm is presented clearly.
2. As far as I know, the idea of compress activation using cluster to reduce communication for MoE training is rather original.
3. The authors focus on a timely and important problem, especially because there has been a trend of adopting MoE in large-scale models recently.

**Weaknesses:**

1. To convince the audience of this problem's importance and increase the work's significance, the authors may provide a more detailed analysis of the communication overhead. One example is given in Section 2.2. However, it would be better if the authors could study what factors influence the percentage,  perhaps the relationship between the communication overhead and the scale of the training servers, the scale of models, etc.

2. Lack of background and related work. I would suggest the authors provide background on LSH algorithms, such as how to calculate centroid using LSH. Further,  as far as I know, there are more works that improve the efficiency of MoE training, such as DeepSpeed MoE, DeepSpeed-TED, and SCoMoE, which should be discussed and compared.

**Questions:**

1. Why did T5-MoE achieve a much faster convergence than RoBERTa-MoE?

2. Figure 5 shows a difference in validation perplexity( 2.37 vs. 2.5), while LSH-MoE seems not quite flattened out. Would continuing training LSH-MoE reach the same perplexity?

3. For the pertaining setting, what is the accuracy difference on zero-shot or few-shot downstream tasks between origin and LSH-MoE?
The

**Limitations:**

The authors claim there is no limitations and no societal impact of the work performed.

---

> ### Author Rebuttal · Authors · 2024-08-07
>
> # W1
> Thank you for your suggestion to deepen our analysis of communication overhead.
>
> To address the review's comment, we have conducted a detailed deduction in the global response to analyze the communication overhead. In particular, the ratio of communication time to computation time can be formalized as:
>
> $\frac{T_{all-to-all}}{T_{compute}} = \frac{4 l \frac{nk}{w} \times \frac{h (w-1)}{B_{inter}}}{(96 n l h^2) / \text{FLOPs}} = \frac{k \times \text{FLOPs}}{24B_{inter}} \times \frac{w-1}{wh}$
>
> where the first term $\frac{k \times \text{FLOPs}}{24 \times B_{inter}}$ remains constant, $k$ is the number of experts activated per token, and $\text{FLOPs}$ and $B_{inter}$ represent GPU and network performance respectively. Here, $w$ denotes the number of devices, and $h,l$ denote the hidden size and number of layers of the model.
>
> As MoE models scale up, the focus is typically on adding more layers and experts, while the growth in hidden size (i.e., $h$) tends to be gradual. Therefore, the proportion of communication time remains substantial as both model size and server scale increase. This observation underscores the ongoing effectiveness of the LSH-MoE method in larger environments, thereby reinforcing its scalability and future applicability.
>
>
> # W2
>
> Thank you for pointing out the need for more background on LSH algorithms and the missing comparisons with other MoE work about efficiency improvements. In our revised manuscript, we will include a more comprehensive background as well as a related work section.
>
> **Background on LSH algorithms**. Locality-Sensitive Hashing (LSH) is a probabilistic method primarily used to approximate nearest neighbor search in high-dimensional spaces. The key operations in LSH including **Mapping Data into Buckets** and **Calculating Cluster Centroids**. Due to the response word limit, please see the official comment for further details.
>
> **Related Work**. Several advancements have been made to improve the distributed training of MoE models on bandwidth-constrained clusters. DeepSpeed-TED integrates Tensor Parallel, Expert Parallel, and ZeRO-powered Data Parallel to enable training of larger models, and introducing the Duplicate Token Dropping technique to eliminate unnecessary communications. DeepSpeed-MoE introduces a novel MoE architecture, PR-MoE, which selects one expert combined with a shared expert instead of the top-2, thus halving the all-to-all communication volume.
> SCoMoE addresses all-to-all communication in a structured manner, rather than uniformly across different devices. It partitions the data along the sequence or feature dimension and controls the data volume communicated  at different network levels. Furthermore, SCoMoE proposes a token clustering approach that aggregates related tokens before the SCoMoE layers to alleviate routing locality in the structured communication.
>
> However, none of these works consider reducing the All-to-All communication volume in MoE training by compressing the forward activations. Therefore, they can be intergrated with the idea in LSH-MoE for further improvement.
>
> # Q1
> The faster convergence of the T5-MoE model compared to RoBERTa-MoE is mainly due to differences in hardware setups and training tasks. We trained the T5-MoE on an A100 cluster, which offers a computational power of 312 TFLOPs, significantly higher than the 125 TFLOPs of the V100 cluster used for RoBERTa-MoE. Additionally, T5-MoE was trained on a language modeling task, while RoBERTa-MoE used a masked language modeling task. These factors contributed to the observed faster convergence of the T5-MoE model.
>
> # Q2
> We apologize for any confusion caused by the unclear graphical representation in our results. In fact, both models converged at a perplexity of 2.37. However, because the LSH-MoE was plotted with a dashed line, only a single point at 2.37 was visible, which did not clearly convey this outcome. (Please zoom in the right sub-figure of Figure 5 of our manuscript and the single point at 2.37 can be found.)  To fix this issue, we have redrawn the figure in the PDF file attached to our global response (Figure C).
>
> # Q3
> We appreciate your feedback. To demonstrate the generalization capabilities of our MoE models trained with LSH-MoE, we compared their zero-shot performance on the GLUE benchmark. The results, summarized in Figure E of the attached PDF, show that the T5-MoE models trained with LSH-MoE achieved accuracy comparable to standard T5 models, confirming LSH-MoE’s efficacy in pretraining. Because the limited number of tokens in the pre-trained dataset and its out-of-domain nature compared to the GLUE evaluation data, the zero-shot performance metrics are relatively low.
>
>
> # Limitations
> We acknowledge the oversight that our initial submission did not explicitly address potential limitations and societal impacts. At the current stage, our work only considers MoE models. Nevertheless, we want to clarify that MoE models are also a mainstream class of deep learning models that are increasingly adopted due to rising model computational demands and training costs, such as  Mixtral-7Bx8MoE, DeepSeek-MoE, and GPT-4. Hence, accelerating MoE training is indeed a critical direction. Additionally, the core of our work leverages data redundancy, which is also present in non-MoE model training. We hope our observations and utilization of data redundancy can inspire more refined work in optimizing training for non-MoE models as well. We will include a section in our final version of the paper.

---

> ### Author Response · Authors · 2024-08-07
> **Background on LSH algorithms**
>
> # **Background on LSH algorithms**.
> Locality-Sensitive Hashing (LSH) is a probabilistic method primarily used to approximate nearest neighbor search in high-dimensional spaces, which reduces the dimensionality of data by mapping similar data to the same "buckets" with high probability using hash functions. This approach contrasts with traditional exhaustive search methods, offering a substantial reduction in computational complexity, particularly beneficial for large-scale data applications.The key operations in LSH including:
>    + **Mapping Data into Buckets**: At the core of LSH is a family of hash functions that maximize the probability of nearby points in the original space staying close in the hashed space, while distant points are likely to end up in different buckets. Each hash function $h$ is characterized by the property: $P[h(x) = h(y)] = 1 - d(x, y)/D$, where $d(x, y)$ is the distance between points $x$ and $y$ and $D$ denotes the diameter of the space.
>    To map similiar data into the same bucket, multiple hash functions from this family are selected based on the specific attributes of the data (e.g., Euclidean distance, cosine similarity) and the desired granularity of the buckets. Data points are then hashed by these functions, and each point is assigned to buckets according to its hash values, effectively categorizing similar items together for clustering.
>    + **Calculating Cluster Centroids**: By grouping data points into buckets as determined by their hash values, data points are effectively clustered. Each bucket represents a cluster of data points and the centroid of each cluster is then calculated as the mean of all points within that cluster, formulated as: $C_j = \frac{1}{n_j}\sum^{n_j}_{i=1}{x_i}$, where $C_j$ is the centroid of the j-th bucket, ​$n_j$ is the number of points in the j-th bucket, and $x_i$ are the data points in the bucket.

---

> > ### Comment · Reviewer_4XaA · 2024-08-12
> >
> > Thanks for the reply. I will increase my score, as the authors clarify most of my concerns. And I would encourage the authors to include the update in the final manuscript.

---

> > > ### Author Response · Authors · 2024-08-13
> > >
> > > Dear Reviewer 4XaA,
> > >
> > > Thank you for your positive feedback and for increasing the score of our paper. We are grateful for your encouragement and will ensure that the suggested updates are included in the final manuscript.
> > >
> > > Best regards,
> > >
> > > Authors of Paper 4459

---

### Official Review · Reviewer_ooKw · 2024-07-15

**Soundness:** 3
**Presentation:** 3
**Contribution:** 3
**Rating:** 6
**Confidence:** 4

**Summary:**

The paper introduces LSH-MoE, a communication-efficient training framework for Mixture-of-Experts (MoE) models using Locality-Sensitive Hashing (LSH). The authors identify the inefficiencies in existing MoE training methods, particularly the high communication costs due to all-to-all communications among GPUs. The proposed method leverages token similarity for data compression, significantly reducing communication overhead and achieving substantial speedups in training time while maintaining model quality.

**Strengths:**

1. The use of Locality-Sensitive Hashing for compressing communication data in MoE training is novel and addresses a significant bottleneck in distributed training systems.
2. The authors conduct extensive experiments on various language and vision models, demonstrating the effectiveness of their method across different tasks and datasets.
3. The experimental results show impressive speedups (1.28-2.2×) in training time, making the approach highly beneficial for large-scale model training.

**Weaknesses:**

Maybe it would be better to include more larger MoE results.

**Questions:**

How does the authors view the potential speedup of LSH-MOE over larger MoE models like Mixtral 8x7B?

---

> ### Author Rebuttal · Authors · 2024-08-07
>
> We would like to address the reviewer's concerns about the speedup over larger MoE models with both experiments and analysis.
>
> **Experiments**
>
> We conducted experiments on the GPT-MoE 52B model, and the results are summarized in Fig. E of the one-page PDF file attached with our global response.
> The results demonstrate that as the parameter size of the MoE models increases, LSH-MoE continues to achieve significant improvements without comprising model accuracy.
>
> Other larger MoE models, such as Mixtral 8x7B, operate on the same mechanisms as GPT-MoE. Unfortunately, due to the time constraint of rebuttal, we were unable to finish the coding development and coordination of experimental resources for experiments on the Mixtral 8x7B model. However, we believe that the results on the GPT-MoE 52B model sufficiently demonstrate the applicability and effectiveness of our method on larger models
>
> **Analysis**
>
> As analyzed in our global response, the ratio of communication time to computation time can be formalized as:
>
> $\frac{T_{all-to-all}}{T_{compute}} = \frac{k \times FLOPs}{24 B_{inter}} \frac{w-1}{wh}$
>
> where $k$ is number of experts activated per token, $FLOPs$ and $B_{inter}$ represent GPU and network performance, $w$ is the number of devices, and $h$ is the hidden size of model.
>
> We have the two facts that
> - the first term $(k \times FLOPs)/(24B_{inter})$ is constant;
> - scaling MoE models often focus on increasing the number of layers and the number of experts, while the growth in hidden size (i.e., $h$) tends to be gradual, e.g., Switch-Transformer.
>
> Therefore, when both the scale of models and the scale of training servers expand, the proportion of all-to-all communication time remains nearly constant. This observation supports the continued efficacy of the LSH-MoE method in larger settings, thereby affirming its scalability and potential in future applications.

---

### Official Review · Reviewer_7fVV · 2024-07-26

**Soundness:** 3
**Presentation:** 3
**Contribution:** 3
**Rating:** 7
**Confidence:** 4

**Summary:**

This paper presents a method to speed up MoE large model training with locality-sensitive hashing. It conducts experiments on both language models and vision models for both pre-training and fine-tuning tasks, and achieves 1.28-2.2× of speedup.

**Strengths:**

1. This paper introduces an efficient LSH-based compression technique to apply tokens on similar expert model, which is both effective and fast.
2. This paper is well written for understanding.
3. This paper conducts experiment on different types of large model and training tasks, which shows the generality of proposed method.

**Weaknesses:**

1. There is a lack of the whole framework figure for the proposed LSH-MoE method for better understanding of reader.
2. More deeper analisis should be added for  ablation study,  the experiment result shows the effect of different quantity and type of hash functions. But we still don't have even a guess of reason.
3. Some expressions need to be improved for example, what is the difference between compression rates and compression ratio in Fig 6.

**Questions:**

1. What's the difference between compression rates and compression ratio in Fig 6, and why we choose 20%,15%,10% compression rates for accuracy comparation.
2. We only try  cross-polytope and spherical hashing for comparation, are there any other hashing types could be adopted?
3. Do we consider  to adopt some learning to hash method for MoE model training speedup? Because as far as I see, learning to hash method usually keep semantic similarity better than LSH method.

**Limitations:**

The authors have not  addressed the limitations and negative societal impact of their work. This work gives a good training speedup solution for MoE large models. But the proposed method could not work well on none-MoE models. Maybe the author can consider some speedup method for general large models.

---

> ### Author Rebuttal · Authors · 2024-08-07
>
> # W1
> To address the reviewer's comment, in the attached PDF file of our global response, we have included a figure (Fig A) to depict the schematic of MoE Training with Locality-Sensitive Hashing (LSH-MoE).
> This figure highlights the key components of LSH-MoE, including the *LSH-Based Clustering* and the *Residual-based Error Compensation*. These components play a significant role in leveraging data redundancy to accelerate the training process.
>
> As illustrated in Fig A, LSH-MoE initially employs (1) an LSH-Based Clustering method to compress tokens into centriods for subsequent processing, effectively reducing communication cost. It then sequentially executes (2) all-to-all communication, expert computation, and another (3) all-to-all communication to produce the processed outputs E(centriods). Finally, it introduces (4) a Residual-based Error Compensation method to approximate the expert-processed results E(tokens), by integrating E(centriods) with residuals.
> # W3 & Q1
> We use the terms "compression rate" and "compression ratio" interchangeably to refer to the ratio of the data volume after compression to that before compression, where a lower value indicates more effective compression.
> We apologize for any confusion due to typographical errors. We will carefully revise our manuscript.
> # W2 & Q1
> There are two key hyperparameters in LSH to reduce hash collisions and optimize model performance, which are the type and quantity of hash functions. Thus, we carried out two kinds of ablation studies in Section 4.5 (Fig 6) of our submitted manuscript, aiming to provide guidance on hyperparameter selection:
>
> **Impact of the Quantity of Hash Functions**: We controlled the number of hash functions to indirectly adjust the LSH compression rate, exploring its effect on model performance across different settings. Specifically, we utilized 2, 4, 6, 8, and 10 hash functions, observing that an increased number of buckets enhances data distinction and, consequently, the compression rate. Results are shown in the left and middle subfigure of Fig 6, where we see improved model convergence quality and worse compression ratio with more hash functions. Importantly, our results indicate that a compression rate of approximately 20% (achieved with about 6 hash functions) is optimal for maintaining nearly identical convergence as uncompressed models, without significantly slowing down training. As described by Zipf's Law [1], there is an imbalance in data distribution (in the corpus of natural language, the frequency of a word's occurrence is inversely proportional to its rank in the frequency table). Therefore, most of the data will be repetitive, leading to data redundancy.
>
> **Impact of the Types of Hash Functions**: We further explored the impact of the types of hash functions with Cross-Polytope Hashing (CP) and Spherical-Plane Hashing (SP).
> As shown in the right sub-graph in Fig 6, CP generally achieves better convergence than SP at the same compression rate. This is attributable to CP's ability to more effectively handle a variety of complex data patterns.
> CP encodes data based on an n-dimensional cross polytope, while SP relies on the geometric relationships between spheres and planes. Thus, CP is more generalizable across a variety of complex data patterns while SP performs better with data that has spherical distribution characteristics. Other works (e.g. Reformer [2]) also use CP to leverage the sparsity of attention mechanisms.
>
> # Q2 & Q3
> Indeed, other hash types can be considered, such as Random Projection Hashing, MinHash, and SimHash. We chose Cross-Polytope Hashing and Spherical-Plane Hashing in this work because they are commonly used in deep learning [2,3,4].
>
> It’s an interesting topic to explore how to integrate our work with learning-to-hash techniques, which often preserve semantic similarity more effectively than tranditional LSH methods.
> However, as model parameters are continuously updated during training, the hidden states also change accordingly, so it necessitates dynamically adjusting learning-to-hash strategies to adapt to the dynamic data distribution, which introduces significant time overhead and affects training efficiency.
>
> Although it is important and challenging to determine the optimal hash functions (such as exploring other hash types or learning to hash), we would like to highlight that this does not hamper the significance of our work. In particular, the core contribution of our work is the identification of data redundancy during the training process and designing a framework that utilizes the LSH method to compress the AllToAll communication volume in MoE training. This allows us to accelerate model training while ensuring model convergence accuracy. We believe that our work will spark further interest in data redundancy and inspire more research in this area. Consequently, we would like to leave the investigation of the optimal hash functions as a potential future direction.
>
> # Limitation
> Thank you for raising this concern, and we acknowledge it as valid.
> Meanwhile, we want to clarify that MoE models are also a mainstream class of deep learning models that are increasingly adopted due to rising model computational demands and training costs. Examples include Switch-Transformer, Mixtral-7Bx8MoE, DeepSeek-MoE, and possibly models like GPT-4.
> Therefore, we believe accelerating MoE training is indeed a critical direction and our work is of significant value.
>
> [1] Zipf's law. https://en.wikipedia.org/wiki/Zipf%27s_law
>
> [2] Kitaev N, Kaiser Ł, Levskaya A. Reformer: The efficient transformer. ICLR 2020.
>
> [3] Bojarski M, Choromanska A, Choromanski K, et al. Structured adaptive and random spinners for fast machine learning computations. PMLR 2017.
>
> [4] Dahlgaard S, Knudsen M, Thorup M. Practical hash functions for similarity estimation and dimensionality reduction. NeurIPS 2017.

---

> > ### Comment · Reviewer_7fVV · 2024-08-13
> > **Rating after reading the rebuttal**
> >
> > I have read the rebuttal，and I would like to keep my rating for this paper.

---

> ### Author Response · Authors · 2024-08-13
>
> Dear Reviewer 7fVV,
>
> Thank you for taking the time to read our rebuttal and for maintaining your positive rating of our paper. We greatly appreciate your support and confidence in our work.
>
> Best regards,
>
> Authors of Paper 4459

---

### Author Rebuttal · Authors · 2024-08-07

We sincerely appreciate the reviewers for their thorough evaluations and valuable feedback.

In the attached PDF, we have provided a detailed response to each comment from all reviewers. To aid in understanding our responses, we have included the figures recommended by the reviewers.We hope that this detailed reply adequately addresses the issues raised and demonstrates our commitment to improving the quality of our work.

To demonstrate the scalability of our work (its applicability to larger models and larger clusters), we theoretically derived the ratio of communication time to computation time as the machine scale and model scale increase.

First of all, we want to highlight that as the scale of both models and machines increases, the proportion of all-to-all communication time relative to the total time remains nearly constant. This consistency suggests that the LSH-MoE method remains effective even at larger scales. We will now present our derivation step by step, using the following notations:

+ $n$: number of tokens processed per GPU
+ $m$: number of tokens communicated between two training servers
+ $k$: number of experts activated per token
+ $h$: hidden size for each token
+ $l$: number of layers for the model
+ $w$: number of training servers
+ $B_{intra}$: the intra-machine network bandwidth
+ $B_{inter}$: the inter-machine network bandwidth


# 1. Formulate all-to-all communication
For any given training server, the amount of tokens (i.e., $m$) communicated with any other GPU node can be expressed as $m=n \times k/w$. Similarly, the volume of communication within the same GPU node is also equal to $m=n \times k/w$. Consequently, the time required for all-to-all communication during model training can be modeled as follows, with each layer involving two instances for the forward pass and two for the backward pass:

$T_{all-to-all} = 4  \times  l  \times  (\frac{m \times h}{B_{intra}} + \frac{m \times h \times (w-1)}{B_{inter}}) \approx 4  \times  l  \times  \frac{n \times k}{w} \times \frac{h \times  (w-1)}{B_{inter}}$


# 2. Formulate model computation
Based on the derivation in [1], for a standard decoder model, given the number of layers $l$, and the hidden size $h$ of the model, the parameter count can be formalized as $|Params| = l  \times  16  \times  h^2$. According to the theory in the appendix of the GPT-3 paper [2], the computation time per GPU can be formalized as $T_{compute}$, where FLOPs reprensents the computation ability of GPU.

$T_{compute} = 6 \times |tokens| \times \frac{|Params|}{FLOPs} = \frac{96  \times  n  \times  l  \times  h^2}{FLOPs}$


# 3. Formulate all-to-all communication/computation
Therefore, as the machine scale ($w$) and model scale ($l$ and $h$) increase, the ratio of computation time to communication time can be formalized as:

$\frac{T_{all-to-all}}{T_{compute}} = \frac{4  \times  l  \times  \frac{n \times k}{w} \times \frac{h \times  (w-1)}{B_{inter}}}{(96  \times  n  \times  l  \times  h^2) / FLOPs} = \frac{k \times FLOPs}{24 \times B_{inter}}  \times  \frac{w-1}{w \times h}$

where the first term $\frac{k \times FLOPs}{24 \times B_{inter}}$ is constant.


# Conclusion
As MoE models scale up, the emphasis is generally placed on increasing the number of layers and experts, with a more gradual increase in hidden size. Consequently, the proportion of communication time remains significant as both the model size and the number of servers increase. These observations and theoretical proofs underscore the sustained effectiveness of the LSH-MoE method in larger environments, thus reinforcing its scalability and applicability for future advancements.

[1]. How to Estimate the Number of Parameters in Transformer models. https://towardsdatascience.com/how-to-estimate-the-number-of-parameters-in-transformer-models-ca0f57d8dff0

[2]. Brown T, Mann B, Ryder N, et al. Language models are few-shot learners. NeurIPS 2020.

[3] Fedus W, Zoph B, Shazeer N. Switch transformers: Scaling to trillion parameter models with simple and efficient sparsity. JMLR 2022.

---

### Decision · Program_Chairs · 2024-09-25

**Decision:**

Accept (poster)

**Comment:**

This paper proposes a communication-efficient method to train MoE models. The method leverages token similarity and LSH to significantly reducing communication overhead while maintaining model quality.

The AC agrees with the reviewers that the problem being studied is important and timely. The proposed method is technically sound. The paper is in general well written. The method is well evaluated showing impressive improvements.

The authors have addressed most of the concerns from the reviewers, and the final scores of the work are unanimously acceptance (7, 6, 6, 7). Please add the additional discussions especially the scalability results in the final version.